# Emerging Roles of Cancer Stem Cells in Bladder Cancer Progression, Tumorigenesis, and Resistance to Chemotherapy: A Potential Therapeutic Target for Bladder Cancer

**DOI:** 10.3390/cells9010235

**Published:** 2020-01-17

**Authors:** Amira Abugomaa, Mohamed Elbadawy, Hideyuki Yamawaki, Tatsuya Usui, Kazuaki Sasaki

**Affiliations:** 1Laboratory of Veterinary Pharmacology, Department of Veterinary Medicine, Faculty of Agriculture, Tokyo University of Agriculture and Technology, 3-5-8 Saiwai-cho, Fuchu, Tokyo 183-8509, Japan; s193249s@st.go.tuat.ac.jp (A.A.); mohamed.elbadawy@fvtm.bu.edu.eg (M.E.); skazuaki@cc.tuat.ac.jp (K.S.); 2Faculty of Veterinary Medicine, Mansoura University, Mansoura 35516, Dakahliya, Egypt; 3Department of Pharmacology, Faculty of Veterinary Medicine, Benha University, Moshtohor, Toukh 13736, Elqaliobiya, Egypt; 4Laboratory of Veterinary Pharmacology, School of Veterinary Medicine, Kitasato University, Towada, Aomori 034-8628, Japan; yamawaki@vmas.kitasato-u.ac.jp

**Keywords:** bladder cancer, cancer stem cells, drug resistance, organoid, molecular targeting therapy

## Abstract

Bladder cancer (BC) is a complex and highly heterogeneous stem cell disease associated with high morbidity and mortality rates if it is not treated properly. Early diagnosis with personalized therapy and regular follow-up are the keys to a successful outcome. Cancer stem cells (CSCs) are the leading power behind tumor growth, with the ability of self-renewal, metastasis, and resistance to conventional chemotherapy. The fast-developing CSC field with robust genome-wide screening methods has found a platform for establishing more reliable therapies to target tumor-initiating cell populations. However, the high heterogeneity of the CSCs in BC disease remains a large issue. Therefore, in the present review, we discuss the various types of bladder CSC heterogeneity, important regulatory pathways, roles in tumor progression and tumorigenesis, and the experimental culture models. Finally, we describe the current stem cell-based therapies for BC disease.

## 1. Introduction

Bladder cancer (BC), referred to as urothelial carcinoma (UC), is the most frequent neoplasm of the urinary tract. BC is associated with high morbidity, mortality, and high costs for treatment [1,2]. It is the fifth most occurring cancer in the United States; however, the laboratory models that reflect the biology of the disease are scarce. The BC disease is about four times more frequent in men than in women with comparable mortality, implying that women are prone to have more aggressive forms of the disease [1], likely due to the signaling pathway convergence. Most human BC patients are the non-muscle invasive (NMI) type with a favorable diagnosis [3], while to a lesser extent it is muscle-invasive (MI) with high metastasis and poor prognosis [1]. 

Although BC is frequent, it is often difficult to manage and control. According to morphology, BC can be classified into papillary, solid, and mixed types. The papillary type is predominant, especially in NMIBC [1]. Genetically, BC can be grouped into a basal or luminal subtype [4,5]. The basal subtype of BC is more complicated, difficult to treat, shows more stemness and epithelial-mesenchymal transition (EMT) [5], and is often metastatic [6] more than the luminal subtype which is mostly nonmuscle-invasive [5,6]. The distinct clinical consequences and aggressiveness of BC differ according to its molecular profiles [7,8]. Most low-grade NMIBC showed mutation of fibroblast growth factor receptor 3 (FGFR3) with the worst outcomes noticed in patients with TP53 and ERBB2 (HER2) mutations [9], while the majority of the advanced grade of MIBC revealed a loss of TP53 function [10].

Urothelial carcinoma could be regarded as a stem cell disease. Analyses on the molecular signature of BC stem cells revealed heterogeneity and intrinsic plasticity, which markedly influences their response to therapy. Therefore, having a good understanding about the stemness of BC is a prerequisite to improving the treatment of this disease. In this review, we describe cancer stem cells (CSCs) in BC disease, their important markers, and their roles. Additionally, we introduce different experimental culture models and newly developed stem cell-based therapy for BC disease.

## 2. Stem Cells in Normal and Tumor Bladder Tissues

Physiologically, the normal stem cells are located in the basal cell layer of the urothelium to maintain homeostasis, renewal, and integrity of the urothelium after damage [11]. Many markers are expressed, including CD44, CK5, CK17, and laminin receptors [12]. In order to identify and target tumor-initiating cells, the analysis of normal cells and CSCs from the same tissues has been employed and revealed that several markers have been found in their malignant counterparts [11]. Among them is OCT4, a key regulator of self-renewal embryonic stem cell markers, which shows high expression in human BC. OCT4 is also associated with its high progression rate and aggressiveness [13]. Another marker is CD44, a prominent stem cell marker located in the basal cell layer of the normal and tumor urothelium [14].

CSCs are tumor-initiating clonogenic cells, which are capable of conserving cellular heterogeneity, self-renewal, and differentiation [15], and they drive the tumor growth, metastasis, and resistance to conventional anti-cancer drugs [16,17]. It is widely assumed that CSCs may arise from normal stem cells that underwent gene mutations [18] via complex mechanisms [19]. Also, the normal urothelial stem cells and differentiated basal cells, intermediate cells, and umbrella cells can acquire tumorigenic potentials and transform into CSCs [11,20]. Identifying predictive markers that have crucial roles in the management of BC helps with better management of this disease. Several CSC surface markers have been identified as responsible for BC development, progression, maintenance of stemness, metastasis, and recurrence [21]. Among them are CD44, CD67LR, EMA, CD133, SOX2, SOX4, ALDH1A1, EZH1, BMI1, MAGE-A3, PD-L1, YAP1, and COX2/PGE2/STAT3, as well as the molecules related to hedgehog, phosphoinositide 3-kinase (PI3K)/AKT, Wnt/β-catenin, Notch [21,22], and c-Myc signaling pathways [23,24].

## 3. Roles of CSC Markers in BC Progression and Tumorigenicity

Clinically, identifying reliable prognostic markers to characterize if the NMI type of BC is more prone to develop than MI type is still missing, and the use of CSC markers of BC as a prognostic tool has been limited [16]. The potential diagnostic and therapeutic CSC markers reported in many papers are introduced below (Figure 1).

### 3.1. Role of CD44

CD44 is a widely expressed cell surface adhesion molecule that is overexpressed in several neoplasms and involved in cancer cell proliferation, differentiation, migration, angiogenesis, and disease progression [25]. CD44^+^ cells are located in the basal layer of the normal urothelium [26,27]. Chan et al. found that the ability of CD44^+^ tumor cells to form a tumor in immunodeficient mice was 10–200 times higher than CD44^−^ tumor cells, and the expression analysis of CD44 in a tissue array of over 300 BC showed that the subpopulation of CD44^+^ cells comprised around 40% of all tumor cells [28]. 

Another study [29] has shown similar CSC subsets and examined the possibility of EMA^−^ CD44v6^+^ as molecular markers of BC-initiating cells. The EMA^−^ CD44v6^+^ cell population represents about 30% of the total cancer cell population. An in vitro single-cell cloning assay revealed that CD44^+^ cells have a high self-renewal ability and the same clonogenic capacity as the parental tumor [29]. Additionally, overexpression of specific CD44 RNA splicing variants, namely CD44v8-10 in BC, has been shown to be closely associated with tumor progression, aggressiveness, and metastasis [30,31]. Its ratio in BC tissue and urinary exfoliated cells showed a significant linear correlation in the same patients. It is highly expressed in BC tissue especially in patients with MIBC versus NMIBC. Overexpression of CD44v9 in BC was also associated with EMT-mediated invasion and migration and a worse prognosis [32].

The main ligand for CD44 is hyaluronic acid (HA), a major component of the extracellular matrix that is expressed by stromal and tumor cells [33]. The HA binds the CD44 ligand-binding domain causing conformational alterations that, in turn, activate several signaling pathways, yielding cell proliferation, adhesion, migration, and invasion [34,35]. Suppression of CD44-HA crosstalk results in the annulment of tumor cell motility [36]. In addition, inhibition of the hyaluronic synthases1, CD44v3, CD44v6, and CD44s resulted in the abolishment of BC growth, invasion, and angiogenesis [37]. Collectively, CD44 may serve as a good prognostic marker of BC [30,38].

### 3.2. Role of ALDH1A1

The ALDH1A1-positive cell population is enriched in BC and its upregulation is associated with progression, aggressiveness, recurrence, shorter survival time, and a poor prognosis of BC [39,40]. Moreover, knockdown of the ALDH1A1 gene by the specific siRNA significantly decreased its clonogenicity and tumorigenic potential in bladder CSCs [39,41]. It is one of the top upregulated genes of BC organoids compared with normal bladder tissues [27]. Su et al. found that ALDH1A1^+^ BC cells have higher clonogenicity and tumorigenicity than ALDH1A1 ones [39]. Recently, ALDH1A1 in patient-derived BC spheroids was shown to stimulate retinoic acid signaling, which leads to the overexpression of tubulin b III (TUBB3) and BC progression [41]. 

These data suggest that the ALDH1A1^+^ cells might be enriched in CSCs of BC. Clinically, Orywal et al. showed that a significantly higher total activity of ALDH was found in the serum of both low-grade and high-grade BC patients with high sensitivity and specificity [42,43]. Also, the detection of ALDH1A1 expression using immunohistochemical staining is very simple and easily performed clinically, as its antibody can specifically bind its target with very low or no background [39]. Therefore, ALDH1A1 could potentially be a prognostic marker to improve the accuracy of clinical outcomes and the selection of accurate therapy for BC in patients with disease recurrence and progression.

### 3.3. Role of SRY-Related HMG-Box4 (SOX4)

SOX4 gene overexpression has been reported in various human cancers including BC [44,45] and is associated with tumorigenesis and cancer progression through the EMT [46]. A recent study found that SOX4 regulates BC cells by the repression of WNT5a [47]. Shen et al. [40] showed the importance of SOX4 in the regulation of bladder CSC properties, and its overexpression was correlated with advanced cancer stages and a poor survival rate. They also found that the downregulation of SOX4 results in the inhibition of cell migration, colony formation, and mesenchymal-to-epithelial transition. 

Additionally, knockdown of the SOX4 gene reduced the sphere formation but increased cell populations carrying high levels of ALDH and tumorigenic potentials [40]. Moreover, as a transcription factor, the knockdown of SOX4 was shown to downregulate the cell cycle-related genes, such as CCND1, CDK1, FGFR1, FGFR3, MYB, and MYC, while the tight junction proteins, including CRB3, TJP1, and TJP3 were specifically upregulated [40]. Collectively, these data show that SOX4 is an important regulator of the bladder CSC properties and it may become a useful diagnostic biomarker of the aggressive phenotype of BC.

### 3.4. Role of YAP1

Yes-associated protein (YAP)1, a principle nuclear effector of the Hippo tumor suppressor pathway [48] promotes organ growth in normal tissues and mainly contributes to progression and poor prognosis of BC [49]. Increasing evidence suggests that YAP1 has various roles in tumor growth, immunosuppression, and chemoresistance [50]. Although all underlying mechanisms are not clearly determined, it is evident that activating YAP1 pathways in different cellular components induces an immunosuppressive tumor microenvironment. YAP1 was shown to induce BC cell growth and invasion via the Hippo signaling pathway targeted genes [51]. 

Another study showed that MASK2, a cofactor of YAP1, is necessary for YAP1 to promote BC growth and migration [52]. Ooki et al. observed that YAP1 and COX2/PGE2 pathways interact together for the propagation of bladder CSCs, and that their inhibitors successfully stopped the progression of BC [53]. Moreover, YAP1 has the capability to reprogram the non-CSCs into cells that have CSC-like features [54] and to keep the stemness of tumor cells through the induction of autophagy [55]. These reports suggest the important roles of YAP1 in bladder CSCs and in the growth and progression of BC.

### 3.5. Role of OCT4

OCT4 (also known as POU5f1) is one of the key regulators of self-renewal in embryonic stem cells and its overexpression is potentially associated with tumorigenesis, tumor recurrence, and resistance to therapies. In BC, Atlasi et al. showed that expression of OCT4 was detected in the majority of bladder tumor tissue samples (31/32). However, in the non-neoplastic specimens, it was low (6/22) [13]. Higher OCT4 expression in BC was shown to be linked with the higher grade of human BC and its recurrence after surgery [56]. Moreover, a recent study on 39 BC tissue samples in India revealed that OCT4 expression was correlated with gender, tumor grade, tumor stage, lymph node status, recurrence, progression, and treatment modality [57]. These data indicate that OCT4 could be a valuable clinical diagnostic and prognostic marker for the progression of BC and may become an attractive therapeutic target to establish new therapies for urothelial carcinoma.

## 4. Stemness Variance between NMIBC and MIBC

After molecular characterization of NMIBC and MIBC, stem cell heterogeneity gradually became obvious. The MIBC accounting for 20–30% of human BC is mostly related to the loss of p53, RB, PTEN, and the activation of EMT-related transcription factors [11], and shows higher expression of basal CSC markers, including CD44, P-cadherin, CK5, and CK14 [6,58,59]. However, the NMIBC accounting for 70–80% of human BC cases is commonly correlated with active mutations of FGFR3 and HRAS and reveals abundant expression of totally diverse stemness genes, such as ALDH1A1, ALDH1A2, CD133, Nestin, CD90 [9], the umbrella cell marker Uroplakin 3, and the cell adhesion proteins (LAMB3 and ITGB4) [26]. 

Teixeira et al. [60] observed that some cell populations of MIBC have molecular stemness features with aggressive, high chemoresistance, and tumor-initiating capacities. They used various stem cell-related markers, such as embryonic transcription factors (OCT4, SOX2, and NANOG), ABC transporters (ABCB1 and ABCG2), aldehyde dehydrogenase isoforms (ALDH1A1, ALDH2, and ALDH7A1), and basal urothelial stem cell markers (CD44, CD47, and KRT14). They showed a significant co-upregulation of CD44 and basal-type KRT14, which is correlated with tumor recurrence and short survival rates [61]. Furthermore, they analyzed gene expression patterns in primary tumor specimens and identified a two-gene stem-like signature (SOX2 and ALDH2) potentially useful to characterize MIBC that are more prone to progression or metastasis [60]. 

The above data suggest the leading power of CSCs in the development, progression, and recurrence of MIBC mainly depends on the expression of several stemness-related markers. This supports the interest in discovering new therapeutic strategies that take CSCs into account as target populations and could help practitioners in the future identify BC-diseased patients that could take advantage of more effective therapeutic approaches targeting CSCs at earlier time points.

## 5. Differences of BC Stemness between Males and Females

Another face of bladder CSC heterogeneity is provided by the sexual dimorphism of the disease. As mentioned above, the occurrence of BC is much higher in men compared to women, with a worse overall survival rate in women, suggesting that women tend to have a more aggressive disease [1,62,63]. Although MIBC occurred more frequently in men than in women, the women having MIBC showed a lower survival rate [62]. In men, the primary tumors were more aggressive and tumor recurrences were more invasive [62]. This might be attributed to signaling pathway convergence. 

In mouse models, knockout of the urothelial-specific FOXA1 gene, a transcription factor necessary for activation of Uroplakin II genes, whose loss and/or mutation lead to unfavorable prognosis to BC patients [64,65], resulted in umbrella cell damage, leading to basal cell hyperplasia in male mice and keratinizing squamous metaplasia in female mice. Gene expression analysis revealed upstream of different stemness regulator genes, β-catenin for male-specific basal cell hyperplasia and P63 for female-specific squamous metaplasia [65], suggesting stemness heterogeneity between male and female BC. 

Similarly, the upregulation of urothelial basal cell-specific β-catenin resulted in the development of low-grade NMIBC in males [66]. This could explain the frequency of the disease in males. Thus, androgen-deprivation therapy could be useful to decrease the recurrence rates in male BC, as recently proposed [67,68].

## 6. Regulatory Pathways of Bladder CSCs

Understanding the role of CSCs in BC and their regulatory mechanisms may facilitate therapy and provide a better prognosis. In addition, pathways related to EMT (such as Wnt/β-catenin, Notch, and transforming growth factor-beta (TGF-β) [22]), the sonic hedgehog signaling pathway [69,70], the PI3K/AKT pathway [22], MAPKs [71,72], and the JAK-STAT pathway have been implicated in the development and progression of bladder CSCs [73] (Figure 2).

### 6.1. Wnt/β-Catenin Pathway

The Wnt signaling pathway regulates stemness maintenance, cell proliferation, and cell polarity through its regulation of stem cell homeostasis [74,75]. Wnt signaling is classified into two networks: the canonical Wnt (or β-catenin-dependent); and or non-canonical Wnt (or β-catenin-independent) pathways. In mouse normal bladder tissues, the Wnt/β-catenin pathway was shown to be necessary for the regeneration of the normal urothelium following damage [76,77] and the overexpression of β-catenin under the promoter of Uroplakin II triggered benign hyperplasia in the urothelium of transgenic mice [78]. 

In BC tissues, mutations or genetic changes in the regulatory pathway of Wnt/β-catenin signaling resulted in EMT, the emergence of urothelial CSC phenotype, chemoresistance, enhanced survival, and the promotion of tumorigenesis [79]. In urothelial CSCs, the genotyping of single nucleotide polymorphisms of 40 genes in the Wnt/β-catenin signaling pathway revealed variants in the Wnt/β-catenin stem cell pathway that were proven to have a role in the pathogenesis of BC [80]. Another report also showed that the accumulation of β-catenin and dysregulation of the Wnt/β-catenin pathway induced neoplastic proliferation and increased the potential of invasion in BC [81]. 

The silencing of CpG hypermethylation of the region encoding Wnt inhibitory factor 1 (an inhibitor of the Wnt signaling pathway) repeatedly occurred in BC [82] and the epigenetic silencing of four secreted frizzled receptor proteins, antagonists of the WNT signaling pathway, has been shown as an independent predictor of MIBC [83]. Furthermore, enhancing the expression of Wnt6, by long non-coding RNA UCA1 (urothelial cancer-associated 1), increased the resistance of BC cells to cisplatin and gemcitabine [84,85]. Collectively, all of these data highlight the implication of the aberrant Wnt/β-catenin stem cell pathway mediating the development of BC and the therapies.

### 6.2. Notch Signaling Pathway

The Notch signaling pathway has both oncogenic and tumor-suppressive effects depending on the organ and cellular context [86]. It involves four receptors (Notch1–4) and five ligands and regulates the transcription of multiple target genes [21]. The expression of the Notch1 receptor is decreased in BC and its activation decreases cellular proliferation, suggesting that it has a tumor-suppressive role [86]. On the other hand, overexpression of Notch2 promoted cell proliferation, EMT, invasion, and stemness maintenance, and its inhibition decreased malignant phenotypes in vitro and in vivo [87]. Furthermore, Notch3 overexpression promoted cell growth and chemoresistance in BC [88]. The tested small molecules, GSIs104 and 105 in the clinical trials to inhibit NOTCH signaling [89] showed the disruption of CD44, ERBB4, and cadherin-mediated pathways, leading to varied and unpredictable effects. Due to these opposing roles of NOTCH and the challenges for avoiding side effects in other organs, more evidence will be needed to support the development of NOTCH-targeting therapy in cancer [22].

### 6.3. TGF-β Pathway

TGF-β has been proven to play a central role in cell proliferation, migration, survival, and differentiation [90,91]. TGF-β can trigger the EMT to facilitate tumor cell stemness, invasion, and metastasis [92]. In BC, activation of the TGF-β signaling increased tumorigenesis [93] and promoted EMT through the upregulation of Malat1 expression, which was associated with poor survival in patients with BC [84]. Furthermore, the knockdown of Malat1 inhibited TGF-β-induced EMT and its targeted inhibition suppressed the migration and invasion characteristics triggered by TGF-β [84], suggesting that targeting TGF-β through Malat1 could be a promising therapeutic option for stopping BC progression.

### 6.4. Hedgehog Signaling Pathway

The sonic hedgehog signaling pathway (Shh) has emerged as a critical component of organ development and differentiation during embryonic development. However, upon aberrant activation, it triggers cancer initiation and progression, and resistance to chemotherapy [94,95]. Activation of Shh promoted EMT (through the decreased expression of E-cadherin and ZO-1 and the increased expression of N-cadherin) and subsequently increased invasion, clonogenicity, tumorigenicity, and stemness in BC [70]. Activation of Shh by glycotransferase (GALNT1)-mediated glycosylation and was shown to maintain the self-renewal and tumorigenic potentials of bladder CSCs [96]. Therefore, targeting Shh could be clinically beneficial in reversing the EMT phenotype and could potentially inhibit BC progression and invasion.

### 6.5. PI3K/Akt Pathway

The PI3K/Akt signaling pathway has a crucial role in maintaining cellular proliferation and the biological features of tumor cells such as cellular growth, differentiation, cell cycle, metabolism, survival, apoptosis, angiogenesis, and migration [97]. This pathway is implicated in the resistance to cisplatin chemotherapy [98,99]. PI3K/Akt regulates the process of EMT in different ways to influence tumor aggressiveness, involving activation of integrin-linked kinase activities and a series of relevant transcription factors (Twist, Snail, and Slug) and the stimulation of matrix-degrading proteases (e.g., matrix metalloproteinase). Furthermore, PI3K/Akt might activate the EMT by interacting with other signaling pathways, including nuclear factor-κB, RAS, TGF-β, and Wnt/β-catenin [100]. These data suggest the implication of the PI3K/AKT pathway in BC progression and stemness and the possibility of becoming a potential therapeutic target.

### 6.6. Other Signaling Pathways

Aberrant activation of the Janus kinase/signal transducers and activators for the transcription (JAK-STAT) signaling pathway has been shown to trigger tumorigenesis [73] and regulate stemness and cell survival in cisplatin-resistant BC cells [101]. The activation of STAT3 in bladder CSCs leads to its expansion and unique clinical progression of invasive BC [102]. The phosphorylated STAT3 protein level was upregulated in BC tumor tissues and in six various BC cell lines [103], which was correlated with BC cell growth and survival [104]. Additionally, the KMT1A was shown to positively regulate the self-renewal and tumorigenicity of bladder CSCs in humans via the KMT1A-GATA3-STAT3 circuit, suggesting that KMT1A could be a promising target for BC therapy [105]. These preclinical data suggest that agents and approaches that block the activation of STAT3 in cancer cells could have a substantial additional value in improving the sensitivity to anti-cancer drugs or preventing anti-cancer drug resistance.

## 7. Experimental Culture Models of Bladder CSCs

Cultures of primary mouse and human bladder cells have been reported, but have been limited due to their short lifespan [106,107]. To date, the cultured urinary bladder cell lines represent the most frequently used in vitro model to study the mechanisms of BC development and to evaluate the efficacy of a variety of anti-cancer drugs [108]. The first BC cell line, RT4, was established in 1970 by Rigby and Franks [109] (Figure 3). Later on, several human BC cell lines were developed and characterized according to their origin (muscle-invasive or not), grade (high or low), behavior (metastatic or not), stage (early or late), and genetic profile (p53, PTEN, etc), which are reviewed in DeGraff et al. [110]. 

Rodent BC cell lines were also established from rats [111,112] and mice [113] (Figure 3). All of these, however, fail to recapitulate many aspects of the original tumor and are often difficult to establish. Tumors are three-dimensional (3D) complexes in which there is a great interaction between tumorous and non-tumorous cells. Thus, 3D cancer cell culture was developed (Figure 3). Chang and his team were pioneers in using human BC specimens in 3D histoculture to evaluate the efficacy of a new platinum analog [114]. The developed 3D BC cell culture led to important insights in tumor biology, since features of the in vivo tumor, such as proliferation, signaling mechanisms, chemosensitivity, and nutrient gradients, can be studied under controlled conditions [115]. 

Later on, several studies have shown the value of 3D organoid culture systems for modeling varied aspects of cancer biology [116,117] (Figure 3). The primary BC organoids recapitulated the histopathological and molecular profile of their corresponding parental tumors, retained the tumor heterogeneity and tumorigenicity, and then could be used as a system for examining the sensitivity to chemotherapy. As an in vivo preclinical model, a patient-derived orthotopic mouse model has been recently established and successfully recapitulated clinical urothelial cell carcinoma progression and metastasis [118]. It enables us to investigate the development and metastasis of BC disease. In the future, it may become an essential research model for testing novel anti-cancer therapeutic agents in BC.

## 8. Potential Therapeutic Targets for Bladder CSCs

The high recurrence rates of papillary NMIBC and the high invasions and metastasis of MIBC to regional lymph nodes and distant organs are attributed to the CSCs [26,119], and the recurrent BC shows resistance to most of the chemotherapy [120]. The developed chemoresistance may be due to the difficulty of drugs to obtain access to the core of tumor tissues to eradicate all CSCs, or due to the high stemness of BC. Therefore, the combination of chemotherapy and CSCs-targeted therapy is the most promising protocol for BC therapy. 

Due to the high probability of recurrence of NMIBC and the poor survival rate of MIBC, new therapeutic approaches, like CSCs-based ones, are needed. Although CSCs-based therapy is still far away from the clinical application and still in its infancy, it remains a promising approach to drug resistance and to keep cancer from growing and spreading [121]. Therefore, a comprehensive study on the genomic profiles of CSCs to identify CSCs-related genes in order to determine intervention targets is necessary. The various challenged potential therapeutic options for bladder CSCs are discussed below (Figure 4).

### 8.1. MicroRNAs (miRNAs)

The inhibition of miRNAs involved in CSC functions by the local administration of antagomiRNAs/antimiRNAs or miRNA replacement therapy was shown to reverse the EMT phenotype and kill CSCs in BC by sensitizing them to anti-cancer drugs [122]. In addition, the stable expression of miR-200 in many phenotypically mesenchymal human BC cell lines was shown to regulate EMT and increase the sensitivities of anti-EGFR therapies in BC cells [123]. Furthermore, it downregulated the expression of ERBB receptor feedback inhibitor 1, a novel regulator of EGFR-independent growth (including ZEB1 and ZEB2), upregulated E-cadherin, reduced cell migration, modulated EMT, and induced differentiation [123]. 

A recent study also showed that in BC, direct targeting of WNT5A by miR-374a treatment inhibited the phosphorylation and nuclear translocation of β-catenin, thus decreasing the expression of BC stemness-related proteins and the aggressiveness behavior of BC [124]. Other miRNAs which inhibit bladder CSCs proliferation, self-renewal, and metastasis are miR-139-5p (by targeting the Bmi1 oncogene [125]), miR-379-5p (by targeting MDM2 [126]), miR-125a-5p (by targeting FUT4 [127]), and miR-24 (by targeting CARMA3 [128]), suggesting that miRNAs could become a novel small-molecule therapeutic target for urothelial carcinoma.

### 8.2. Heat Shock Protein (HSP)90 Inhibitors

HSP90 is a molecular chaperone that is necessary for the stability and function of several oncoproteins required for cancer cell evasion, chemoresistance, and self-renewal [129]. The use of 17-DMAG (an HSP90 inhibitor), at non-cytotoxic concentrations, simultaneously inactivated Akt and ERK signaling pathways and synergistically potentiated the cytotoxicity of cisplatin against bladder CSCs by enhancing apoptosis [130]. Ganetespib, a potent inhibitor of HSP90, is currently being used to overcome the therapeutic resistance of CSCs in clinical trials of a wide variety of neoplasms, including BC [131].

### 8.3. Telomerase (TL) Inhibitors

TL activity is upregulated in the majority of human neoplasms [132], tumor cells, and CSCs but not in the adjacent normal tissues [133]. Interestingly, the length of the TL in CSCs is shorter than that of tumor cells [134], suggesting that TL is related to the differentiation of cancer cells. Therefore, targeting TL has been focused on. Hirashima et al. showed that the enforced elongation of TL of cancer cells caused cellular differentiation in vivo [135]. These results suggest the CSCs are sensitive to TL-based therapy. 

Active mutations of the TL reverse transcriptase (TERT) gene and subsequent TL reactivation is often observed in aggressive BC [136]. The C228T mutation of the TERT promoter frequently occurs only in bladder CSCs, not in normal BC cells, which contributes to the tumorigenesis of BC [137]. The potent TL antagonist, GRN163L (a lipid conjugated oligonucleotide N3’→P5’ thio-phosphoramidate), significantly caused growth arrest in T24 BC cells but not in normal urothelial cells [138]. Moreover, the antisense oligodeoxynucleotides against the human TERT reduced the growth of BC cells [139]. These data suggest that TL antagonists may be promising therapeutic agents for bladder CSCs.

### 8.4. Blockade of CD47

Compared with the rest of the BC tumor cells, the CSCs express higher levels of CD47, a cell surface protein that protects CSCs from macrophage phagocytosis [28]. Blockade of CD47 with a monoclonal antibody as a potential type of immunotherapy for BC resulted in macrophage engulfment of BC cells in vitro, suggesting that immunotherapy against CD47 could be effective in the treatment of MIBC.

### 8.5. Anti-PD-1/PD-L1

Recently, Lawrence et al. showed that, like melanoma and non-small cell lung cancer, BC produces tumors with a high somatic mutation frequency and with high antigenic expression [140]. Therefore, BC could be an optimal target for immune-checkpoint inhibitors (ICHs). Although it is still under investigation, the ICHs which target the programmed cell death-1 (PD-1)/programmed death-ligand 1 (PD-L1) pathway or cytotoxic T-lymphocyte-associated protein4 (CTLA-4) [141] have been challenged and shown great promise in advanced BC in some patients.

Shi et al. developed urothelial CSC vaccine (streptavidin-granulocyte-macrophage-colony stimulating factor surface-modified bladder CSCs) which effectively induced specific immune response for eliminating bladder CSCs [142]. In the paper, they also showed that the combination therapy of PD-1 blockade and CSC vaccine therapy elevated the population of CD4^+^, CD8^+^, and CD8^+^IFN-γ^+^ cells, which leads to the enhancement of the functions of tumor-specific T lymphocytes and elimination of bladder CSCs [142].

## 9. Conclusions and Future Perspectives on Bladder CSC Research

Urothelial cell carcinoma of the bladder is considered as a stem cell disease. Identification of early important diagnostics, prognostic markers, and novel therapeutic molecular targets remain the major clinical issues to be addressed in BC. Detailed analyses of the molecular signatures of BC stem cells revealed featured heterogeneity and intrinsic plasticity, which markedly influence the individual response of BC to therapy. The wrong selection of cytotoxic drugs leads to enrichment of tumors with CSCs, which could be the reason for the progressive development of clinical chemoresistance, a major challenge in the treatment of BC. Therefore, a solid understanding of the biological processes involved in the stemness of BC and their correlation with drug resistance is a prerequisite for improving the treatment of this disease.

The treatment patterns for BC continue to expand and there is a positive shift towards more personalized therapy for patients. Identifying specific early diagnostics and prognostic CSC markers in BC may provide a novel method for the classification or clinical staging of the disease, where the level of CSC presence in the cancer tissues may relate to a more aggressive cancer phenotype. Novel CSCs-targeted therapies are still needed for further improving the therapeutic efficacy of BC. Some of the drugs mentioned in this review article have been shown to be promising in the treatment of this disease. Success in therapeutics and the clinical management of BC heavily depends on disease models. Patient-derived BC organoids could represent a faithful model system for studying tumor evolution and treatment response in the context of precision cancer medicine.

## Figures and Tables

**Figure 1 cells-09-00235-f001:**
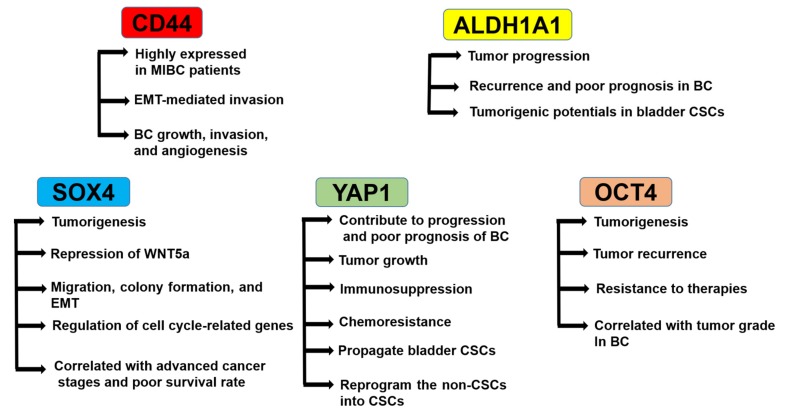
Roles of cancer stem cell (CSC) markers in bladder cancer (BC). CD44 is highly expressed in muscle-invasive (MI)BC patients, regulates epithelial-mesenchymal transition (EMT)-mediated invasion, and mediates BC growth, invasion, and angiogenesis. ALDH1A1 regulates tumor progression, recurrence and poor prognosis in BC, and tumorigenic potentials in bladder CSCs. SOX4 regulates tumorigenesis, repression of WNT5a, migration, colony formation, EMT, and the expression of cell cycle-related genes. The SOX4 expression is correlated with advanced cancer stages and poor survival rates. YAP1 regulates progression and poor prognosis of BC, tumor growth, immunosuppression, chemoresistance, propagation of bladder CSCs, and reprogramming of the non-CSCs into CSCs. OCT4 regulates tumorigenesis, tumor recurrence, and resistance to therapies. OCT4 expression is correlated with the tumor grade in BC.

**Figure 2 cells-09-00235-f002:**
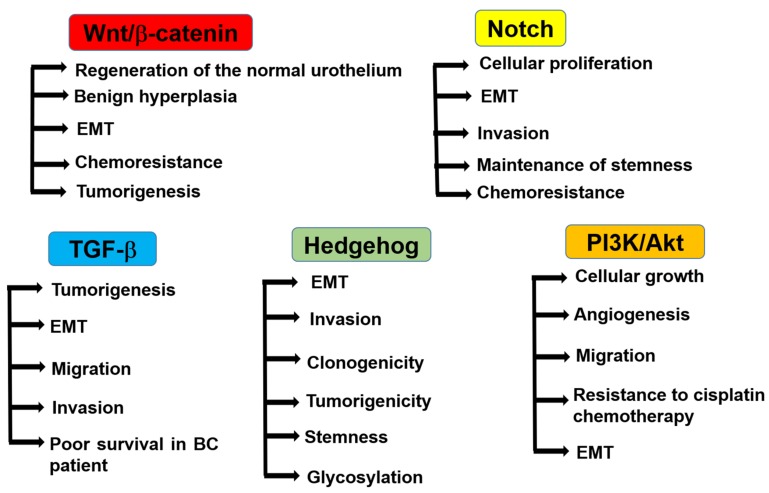
Regulatory pathways of bladder CSCs. Wnt/β-catenin signaling promotes the regeneration of the normal urothelium and benign hyperplasia in normal bladder tissues. In BC, it regulates EMT, chemoresistance, and tumorigenesis. Notch signaling regulates cellular proliferation, EMT, cell invasion, stemness, and chemoresistance in BC. TGF-β signaling regulates tumorigenesis, EMT, and cell migration and invasion in BC. The activation of TGF-β signaling is correlated with poor survival in BC patients. Hedgehog signaling regulates EMT, invasion, clonogenicity, tumorigenicity, stemness, and glycosylation. PI3K/Akt signaling regulates cellular growth, angiogenesis, migration, resistance to cisplatin chemotherapy, and EMT.

**Figure 3 cells-09-00235-f003:**
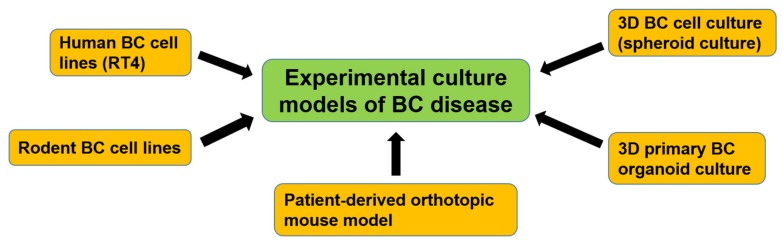
Development of experimental culture models of BC disease. Human and rodent BC cell lines were first developed and used for basic BC research. To study the mechanisms of bladder CSCs, 3D BC cell culture (spheroid culture) was developed. Recently, a 3D primary BC organoid culture method was established, which includes patient-derived bladder CSCs and can mimic the bladder tumor microenvironment. A patient-derived orthotopic mouse model was also established and could recapitulate clinical urothelial cell carcinoma progression and metastasis.

**Figure 4 cells-09-00235-f004:**
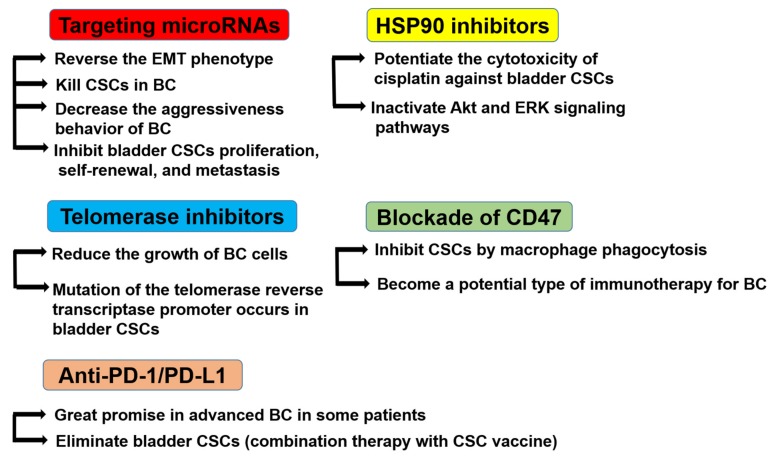
Potential therapeutic targets for bladder CSCs. Targeting microRNAs reverses the EMT phenotype, kills CSCs in BC, decreases the aggressiveness behavior of BC, and inhibits bladder CSC proliferation, self-renewal, and metastasis. HSP90 inhibitors potentiate the cytotoxicity of cisplatin against bladder CSCs and inactivate the Akt and ERK signaling pathways. Telomerase inhibitors reduce the growth of BC cells, and the mutation of the telomerase reverse transcriptase promoter occurs in bladder CSCs. The blockade of CD47 inhibits CSCs by macrophage phagocytosis and may become a potential type of immunotherapy for BC. Anti-programmed cell death (PD)-1/ programmed death-ligand 1 (PD-L1) therapy has shown great promise in advanced BC in certain patients. The combination therapy of PD-1 blockade and CSC vaccine therapy could eliminate bladder CSCs.

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
