# Peer review of "Emerging Roles of Cancer Stem Cells in Bladder Cancer Progression, Tumorigenesis, and Resistance to Chemotherapy: A Potential Therapeutic Target for Bladder Cancer"

_cells, 2020, doi:10.3390/cells9010235_

Round 1
Reviewer 1 Report
1) line114-116: Is the description correct? The reviewer wonders if ALDH1A1+/CD44+ cells may have higher clonogenicity than ALDH1A1+ cells.
2) line133: Did the authors mean EMT, not MET?
3) line171: UPIII would be more specific for umbrella cells than UPI and II.
4) The authors may want to discuss the roles of CSCs in treatment resistance to immunooncology agents (anti-PD1 and PD-L1 Abs), which are currently the standard of treatment for advanced UC.
Author Response
Responses to Reviewer: 1
Q1. line114-116: Is the description correct? The reviewer wonders if ALDH1A1+/CD44+ cells may have higher clonogenicity than ALDH1A1+ cells.
A1. As a reviewer pointed, the ALDH1A1+/CD44+ cells showed similar in vivo tumorigenicity compared with ALDH1A1+ cells (Su et al., Cancer Epidemiol Biomarkers Prev. 19: 327–337, 2010, Table. 1). To avoid confusion, we changed the expression “ALDH1A1+ BC cells have higher clonogenicity and tumorigenicity than CD44+ cells and ALDH1A1+/CD44+ cells’’ to “ALDH1A1+ BC cells have higher clonogenicity and tumorigenicity than ALDH1A1- ones’’ (page 3, line 112-113).
Q2. line133: Did the authors mean EMT, not MET?
A2. We confirmed that it does mean MET.
Q3. line171: UPIII would be more specific for umbrella cells than UPI and II.
A3. Following the suggestions, we changed “Uroplakin 2’’ to “Uroplakin 3’’ (page 5, line 182).
Q4. The authors may want to discuss the roles of CSCs in treatment resistance to immunooncology agents (anti-PD1 and PD-L1 Abs), which are currently the standard of treatment for advanced UC.
A4. Following the suggestions, we added the information on anti-PD-1/PD-L1 therapy against bladder CSCs (page 10, line 397-409, New Figure. 4).

Reviewer 2 Report
To the authors:
The presence of cancer stem cells in bladder cancer could be a biomarker for patient prognosis and potential therapeutic targets. This review paper “Emerging Roles of Cancer Stem Cells in Bladder Cancer Progression, Tumorigenesis, and Resistance to Chemotherapy: A Potential Therapeutic Target for Bladder Cancer” summarized published referenced on this important topic. This is an interesting well written review supported by large number of publications. However, there are some concerns about this manuscript:
Text
Line 38-41: When you talk about percentage prevalence of the grades of the various forms of BCs it seems inconsistent to just say most for MI – High
Line 41: Mentioning total cost of BC therapy as a standalone sentence with nothing leading up to it or following it seems out of place
Line 59-67: From the information given here, OCT4 should be a better marker to target than CD44 since it is only present on CSC not on normal tissue. However, for some reason, the OCT4 is discussed in the rest of the paper.
Line 187-199: Here and earlier when you mention the difference in likely BC mortality as a result of sex it would be prudent to cite studies that specifically back up that there is a higher mortality because of sex as opposed to confounding factors. Without such a source or cite one could pose the argument is it not a function of prevalence and association but not necessarily causation.
Line 207-218 The section on WNT felt light on BC information. It definitely had physiologic information but the WNT mutation BC portion did not necessarily add to the paper as it currently is.
Figures
Most of the figures used (Figures 1, 2, and 4) can be replaced by Tables with checks to the corresponding items that involves the specific mark/pathway. Current figure legends are redundant to what is shown in the figures and the corresponding text. If citations could be added to each point, it would benefit the readers. In addition, consider summarize the paper with a visual diagram indicating the relationship of CSC and non-CSC, pathways involved, impacts, and therapeutic targets.
Figure 3 and its correspondent text only consider the in vitro culture models for CSC study. Preclinical model, especially patient-derived orthotopic mouse models (eg. Gills et al. Oncotarget 2018;9:32718), should also be considered at least in the future directions.
References cited
Format of the citation within text may not be consistent with Journal’s standard. The citation should be [##, ##] if more than one reference is cited. Cite more recent relevant references may add value to the review (see below as a few examples) Some of the citations are not complete, eg, reference number 47, International Journal of Cancer 2019, n/a, (n/a), should be 146(4), 1099-1113.
The conclusion
It neatly summarizes the paper together and provides a level of insight into the organization of information within the paper that the paper itself is lacking. Perhaps edit the paper so that the topics at hand create the message of the paper rather than presenting information.
Author Response
Responses to Reviewer: 2
Q1. Line 38-41: When you talk about percentage prevalence of the grades of the various forms of BCs it seems inconsistent to just say most for MI – High.
A1. Following the suggestions, we revised the description of the explanation of NMI and MI (page 1, line 38-40). Thanks for the comments.
Q2. Line 41: Mentioning total cost of BC therapy as a standalone sentence with nothing leading up to it or following it seems out of place.
A2. Following the suggestions, we deleted this sentence (page 1, line 42).
Q3. Line 59-67: From the information given here, OCT4 should be a better marker to target than CD44 since it is only present on CSC not on normal tissue. However, for some reason, the OCT4 is discussed in the rest of the paper.
A3. Following the suggestions, we added the OCT4 paragraph (Role of OCT4) in chapter 3 (page 4, line 151-161, New Fig. 1).
Q4. Line 187-199: Here and earlier when you mention the difference in likely BC mortality as a result of sex it would be prudent to cite studies that specifically back up that there is a higher mortality because of sex as opposed to confounding factors. Without such a source or cite one could pose the argument is it not a function of prevalence and association but not necessarily causation.
A4. Following the suggestions, we added more information showing the direct relationship between mortality in BC and sex (page 5, line 198-202).
Q5. Line 207-218 The section on WNT felt light on BC information. It definitely had physiologic information but the WNT mutation BC portion did not necessarily add to the paper as it currently is.
A5. Following the suggestions, we added more information on the role of Wnt signaling in bladder CSCs (page 6, line 231-240).
Q6. Most of the figures used (Figures 1, 2, and 4) can be replaced by Tables with checks to the corresponding items that involves the specific mark/pathway. Current figure legends are redundant to what is shown in the figures and the corresponding text. If citations could be added to each point, it would benefit the readers. In addition, consider summarize the paper with a visual diagram indicating the relationship of CSC and non-CSC, pathways involved, impacts, and therapeutic targets.
A6. Thanks for giving us helpful advice. However, we believe that presenting separated Figure style is easier for readers to understand the important points of each chapter compared with Table style including citations. Following the reviewer’s comments, we carefully updated Figures.
Q7. Figure 3 and its correspondent text only consider the in vitro culture models for CSC study. Preclinical model, especially patient-derived orthotopic mouse models (eg. Gills et al. Oncotarget,2018;9:32718), should also be considered at least in the future directions.
A7. Following the suggestions, we added the information on patient-derived orthotopic mouse models (page 8, line 325-329, New Figure. 3).
Q8. Format of the citation within text may not be consistent with Journal’s standard. The citation should be [##, ##] if more than one reference is cited. Cite more recent relevant references may add value to the review (see below as a few examples) Some of the citations are not complete, eg, reference number 47, International Journal of Cancer 2019, n/a, (n/a), should be 146(4), 1099-1113.
A8. Revised (page 15, line 594-597).
Q9. It neatly summarizes the paper together and provides a level of insight into the organization of information within the paper that the paper itself is lacking. Perhaps edit the paper so that the topics at hand create the message of the paper rather than presenting information.
A9. Owing to the reviewer’s helpful advice as above, we believe that we successfully added the proper information and that the revised manuscript became much strengthened. Thanks for giving us many valuable comments.

Round 2
Reviewer 1 Report
The authors adequately addressed the reviewer's comments in the revised version.